# Visualizing the Invisible: Invisible Waste in Diagnostic Imaging

**DOI:** 10.3390/healthcare9121693

**Published:** 2021-12-07

**Authors:** Bjørn Hofmann, Eivind Richter Andersen, Elin Kjelle

**Affiliations:** 1Department of Health Sciences, Faculty of Medicine and Health Sciences, Norwegian University of Science and Technology, P.O. Box 191, N-2802 Gjøvik, Norway; eivind.r.andersen@ntnu.no (E.R.A.); elin.kjelle@ntnu.no (E.K.); 2Centre for Medical Ethics, Institute for Health and Society, Faculty of Medicine, University of Oslo, P.O. Box 1130, N-0318 Oslo, Norway

**Keywords:** waste, imaging, appropriateness, waiting time, overuse, low-value care, quality

## Abstract

There is extensive waste in diagnostic imaging, at the same time as there are long waiting lists. While the problem of waste in diagnostics has been known for a long time, the problem persists. Accordingly, the objective of this study is to investigate various types of waste in imaging and why they are so pervasive and persistent in today’s health services. After a short overview of different conceptions and types of waste in diagnostic imaging (in radiology), we identify two reasons why these types of waste are so difficult to address: (1) they are invisible in the healthcare system and (2) wasteful imaging is driven by strong external forces and internal drivers. Lastly, we present specific measures to address wasteful imaging. Visualizing and identifying the waste in diagnostic imaging and its ingrained drivers is one important way to improve the quality and efficiency of healthcare services.

## 1. Introduction

In the days when X-rays were on film, waste was very visible. Most radiological departments had a box or a bin where all the bad or rejected images were collected. The volume of waste in the bin was a direct and visible indicator of how well the department performed [1]. Today, the bin is gone, but not the waste.

Great geographical, regional and temporal variation in the use of diagnostic imaging [2,3,4,5,6,7,8,9,10] indicates that there is both underuse and overuse of imaging services [11,12,13]. Moreover, 113 ordinary high-volume imaging examinations have been identified as low-value care by the Choosing Wisely Initiative [14]. Additionally, a recent scoping review has identified 84 low-value imaging examinations [15]. Low-value care is a kind of waste that is defined as “use of an intervention in which evidence suggests it confers no or very little benefit for patients, or risk of harm exceeds probable benefit or, more broadly, the added costs of the intervention do not provide proportional added benefits” [16]. Internationally, it is estimated that 20–50% of radiological examinations represent overutilization [17,18].

Paradoxically, at the same time as many healthcare services are providing quite high volumes of low-value imaging, they experience extensive waiting lists with long waiting times for imaging services [19]. If the examinations on the waiting lists were low-value imaging, waiting lists would be a way of priority setting by delay [20]. However, there is little evidence that waiting lists primarily comprise low-value imaging, thus some patients are waiting for high-value imaging because imaging departments are performing low-value examinations.

While this paradox is well known, and a recent systematic review provides an overview of the many interventions available to reduce low-value imaging and their outcomes [21], the problem prevails. Accordingly, the objective of this study is to investigate various types of waste in imaging and why they are so persistent in today’s health services.

The article starts by briefly presenting a short overview of different conceptions and types of waste in diagnostic imaging (in radiology). Then it investigates why these types of waste are so difficult to address. Thirdly, we identify and discuss external forces and internal drivers of wasteful imaging practice. Lastly, we use these findings to present specific measures to address wasteful imaging.

## 2. Waste in Imaging

There are many conceptions of waste in imaging. Table 1 summarizes various definitions of waste identified in the literature on radiology.

More specifically, there are many types of waste in imaging. We will provide a brief overview of waste in imaging using a strategic selection of literature. Hence, the list of waste is not exhaustive. One important type of waste in imaging is deleted images, retakes, rejected images and images without diagnostic value. While it was predicted that retakes would drastically reduce when digital images replaced films (due to improved image production), this has not occurred [29,30]. Retakes and rejects are reported to be at the same level (5–12%) now as when films were used [31,32,33,34,35,36,37,38,39].

Examinations repeated without changes in the patient’s state of health add no clinical utility. These are wasted resources and increase the radiation dose to the patient. This typically occurs when imaging is performed at different centers or hospitals without sharing information of earlier imaging examinations across organizations [40] or if follow-up imaging is ordered too often.

In addition, waste in imaging includes screening examinations not included in evidence-based screening programs. Such examinations may result in inadequate use of resources due to potential overdiagnosis or overtreatment [41]. Furthermore, imaging ordered before clinical examination of the patient, or without relevant information given in the referral, may give a reduced pre-test probability and lowered positive predictive value. This could lead to suboptimal imaging and reporting, retake or delayed patient treatment [42,43].

Table 2 briefly describes the most common types of waste.

The main concern with waste in imaging has been reported to be (1) its extension (too much), (2) its (lack of) utility, (3) patient safety, (4) moral (divergence from norms, professional integrity), (5) deference and (6) lack of control [22]. Hence, the various types of waste raise a wide range of concerns with diagnostic imaging. Moreover, they appear to be persistent despite having been acknowledged and addressed for a long time.

One reason why waste is so persistent in diagnostic imaging is that several forms of waste often are invisible.

## 3. Invisible Waste

Here, we will briefly investigate the various types of waste with respect to how easy they are to detect and measure.

*Retakes*, image rejects, image deletions and images not used for diagnostics are hidden in the computer systems. Where the film bins made waste visible, the non-useful images are hidden at various places in the modality workstation, Picture Archiving and Communication System (PACS), or in the Radiological Information System (RIS). While there exist modules in these systems for reject analysis, these are not frequently used.

*Duplicate orderings* are not visible, as the orders and the imaging are not coordinated and occur at different places.

*Repeated examinations* at too short time interval are not visible to the patient’s physician, the person scheduling and the persons that are performing and interpreting the image.

Imaging is too often performed *without sufficient clinical information*, resulting in poor quality diagnostics. When the image is taken and described, there is (usually) no indication that the image was taken based on too little information.

*Examinations ordered before patient examination* also lack clinical information when the images were taken but this may not be as easy to identify retrospectively. Moreover, those professionals who do the physical examination are not the same as those who take the image. Hence, there are no barriers to imaging before examinations and no feedback to correct this behavior.

In *screening* examinations that are not supported by high-quality evidence, people do not benefit from imaging but may actually be harmed, e.g., by being overdiagnosed and overtreated. Nevertheless, they believe that they have been saved by the screening. This is because it is impossible to differentiate between those who benefit from a positive screening test and those who are *overdiagnosed* (and overtreated) [44]. You cannot tell when you have a positive screening test whether the person will be diseased or die from the condition that you have detected or whether (s) he will die with it.

*Incidental findings of no clinical relevance* are invisible in the same manner as overdiagnosis. While the incidental findings certainly are visible, we may not know whether they are clinically relevant in the individual case. As they are found, we find it difficult to ignore them and tend to inform about them, potentially to act on the basis of them, although the relevance is uncertain or even nil.

In sum, the various kinds of waste in diagnostic imaging tend to be invisible to the various agents. This is one major reason why it is so difficult to reduce waste in imaging. Another reason is that the drivers are so persistent. Here, we will divide the drivers into internal and external.

## 4. External Drivers of Wasteful Imaging

A survey of radiologists revealed that (a) new radiological technology, (b) people’s demands, (c) clinicians’ intolerance of uncertainty, (d) expanded clinical indications and (e) availability were the highest-rated causes of increasing use of radiological investigations in Norway [45]. “Over-investigation” and “insufficient referral information” were reported to be the most frequent causes of unnecessary investigations [45]. Accordingly, a more recent study identified the “need of accuracy of diagnosis,” the “trend of physicians to repeat tests in order to confirm preset diagnoses,” the “lack of knowledge about proper usage of radiological advances” and the “lack of proper clinical examination” as the most common causes of overuse [46].

While most of the drivers of wasteful imaging described above can be classified as external and partly visible drivers, there also appears to be a set of internal and less visible drivers of wasteful imaging.

## 5. Internal Drivers of Wasteful Imaging

Diagnostic imaging is a prominent type of medical technology, and medical technologies are subject to a range of biases and imperatives [47]. Here, we will briefly outline some relevant drivers of wasteful imaging generation:

Strong belief in what you see: Vision is one of our most important senses, and we tend to believe that what we see is real, despite many visual limitations [48]. Medical images imbue a persuasive power, which is based on the natural appearance of their depictions and their assertion of mechanical objectivity [48,49,50]. There is a strong belief that imaging reveals the physical truth. 

Strong belief in advanced technology: There is a strong belief in the benefits of technology in general and in medical technology in particular [47,51]. Advanced technology is considered to be better than simple technologies, and imaging technologies are believed to be amongst the most advanced [47]. Moreover, wasteful imaging also results from increased reliance on technological tests [52], combined with (and partly resulting in) less experience with physical examinations and clinical skills.

To know is better than not to know: Although the proverb “ignorance is bliss” is well known, the opposite is true in the case of imaging. The mentioned strong belief that imaging technologies depict the reality of disease directly [48] makes us believe that having an image taken is always a good thing. We tend to think that we are “better safe than sorry” with an image. When combined, we think that to see is to know, and to know is better than not to know. Altogether, this is related to a fear of uncertainty and to “ambiguity aversion” [53].

Imperative of extension: Moreover, there is a general belief that “more is better than less” [54]. For example, “patients believe that more testing is better” [55]. However, there are many examples where more medicine is not better [22,54,56,57,58].

Aversion asymmetry: In diagnostic imaging, an error of omission appears to be worse than an error of commission [47]. In general, we tend to be more afraid of rejecting a true hypothesis (Type I error) than accepting a false hypothesis (Type II error), i.e., we are more afraid of ignoring than of overdoing.

Defensive medicine and imaging and referrals to protect against litigation may also drive wasteful imaging [59,60,61].

## 6. Discussion

The list of types of waste discussed in this article is not exhaustive. For example, false positive and false negative examination results can be conceived of as waste, as they contribute negatively to providing adequate care. However, they are unavoidable for given imaging procedures.

As many types of waste are invisible, they are difficult to identify, measure and address. For example, deleted images, retakes, rejected images and images without diagnostic value are described as important types of waste, as they are not applied for diagnostic purposes or do not contribute to improved health of the patient in any manner. However, as they are vanishing in the system, their diagnostic value is often not assessed. Thus, they are an obvious wasteful use of healthcare resources, even though a diagnostic value might exist. Moreover, many of these images are related to some (although often small) risks, e.g., with respect to ionizing radiation and contrast agents. Hence, they are not only wasteful, but can also be harmful.

False positives, false negatives and incidental findings are potential consequences of low-value radiology. These are important drivers of further examinations that could be of low value. Hence, the initial use of radiology itself is an important factor for persistent use of low-value examinations in today’s healthcare systems. As technologies are getting better, we are able to find more, leading to more tests and more findings which can be followed up.

Moreover, one important reason for the substantial extension of low-value imaging is the strong belief in imaging and the great demand for imaging services [62]. The general public has a strong belief that medical imaging can depict pathologies directly, and that it gives clear answers to what causes their symptoms. Our compulsion to diagnosis may be one driver of wasteful imaging [63].

On the other hand, knowledge about risks, such as radiation, false positive and negative test results, overdiagnosis, low predictive values and incidental findings (incidentalomas) in the general public are low. Hence, some of the ordinary barriers to waste are diminished. Moreover, many countries have universal coverage, where imaging costs for the individual patients are low. However, even in healthcare systems with out-of-pocket payment, the use of wasteful imaging appears to be high. Thus, the question is when should we stop perusing findings, as more is not always better?

Waste in radiology may have consequences on many levels, from the individual patient and health professional, via the organization of the healthcare system and health policy making to the societal level. For example, increased examination time and a detour on miscellaneous diagnostic tracks may, for the individual patient, be experienced as a substantial disadvantage and physically and mentally draining. For example, incidental findings could lead to anxiety and distress, even though the findings were without clinical consequences. This could further lead to increased use of health services due to follow-ups, further diagnostics and use of support services (such as cognitive therapy), yielding increased strain on the health services.

By visualizing invisible waste in imaging and knowing its drivers, we are better equipped to address this problem. In particular, we must avoid duplicates, reduce retakes, extend time intervals for repeated examinations (where appropriate), halt examinations without sufficient information, stop low-value imaging, including screening with poor evidence to avoid overdiagnosis and overtreatment, as well as being cautious with respect to incidental findings without or with uncertain significance. By reducing waste, we can improve the quality of care by increasing benefit and reducing harm. Moreover, we can promote efficiency and justice, as low-value care resources are freed up to provide high-value care. Many measures for reducing waste are available [21], but it is crucial that they are adapted to the specific context.

This article is not a systematic review, and it is not exhaustive in referencing the literature. However, many references stem from one scoping review [15] and a systematic review [21], which is supplemented with relevant references from related academic fields. Nonetheless, this study can be of great value for further research, including systematic reviews of waste in imaging.

## 7. Conclusions

This article has provided a short overview of different conceptions and types of waste in diagnostic imaging. It then identified what made these types of waste so difficult to detect and address, i.e., that the waste in medical imaging is mostly invisible. Third, we identified and discussed external forces and internal drivers of wasteful imaging, adding to the explanation of why it is so difficult to address wasteful imaging. Lastly, we pointed to specific measures to reduce the waste in imaging in order to improve the quality of care and to promote efficiency and justice.

Hence, while the bin of wasteful film images disappeared, the waste did not. It only became invisible. At the same time, the drivers of wasteful imaging are difficult to address. This can explain the high and persistent level of waste in imaging and the paradox that this exists at the same time as there are extensive waiting times for this very important health service. To reduce waste in imaging, we must avoid duplicates, reduce retakes and repeated examinations, halt examinations without sufficient information, stop low-value examinations and screening with poor evidence, as well as stop looking for incidental findings of uncertain significance.

Hence, visualizing the waste in diagnostic imaging and its ingrained drivers is one important way to improve the quality and efficiency of the healthcare services.

## Figures and Tables

**Table 1 healthcare-09-01693-t001:** Definitions of various concepts used about waste in the literature on radiology. For more details see [22].

Term	Definition
Misuse	“Misuse of radiological tests may exist when they are ineffective (do not affect treatment or outcome) or inefficient.” [23]
Overuse Overutilization	Overuse is “the provision of services to those who are unlikely to benefit”[24]. Overutilization addresses both amount and utility [25]
Unnecessary examinations, use, imaging, X-ray, etc.	Examinations which are “clinically unhelpful in the sense that the probability of obtaining information useful to patients management is extremely low” [26] An unnecessary X-ray is one that “is not going to provide any useful diagnostic information to the physician” [27]
Inappropriate imaging	Tests ”to exclude or rule out disease in people who have only minimal symptoms and a low clinical likelihood of disease, often to reassure both patient and doctor that disease is not present” [23] Inappropriate examinations “include duplicate ordering, absent or nonsupportive clinical information, repeated examinations at an inappropriately short time interval, screening examinations not supported by randomized clinical trial evidence, and examinations ordered before patient examination” [28]
Low-value imaging/care	Low-value care is “an intervention [e.g., an imaging examination] in which evidence suggests it confers no or very little benefit for patients, or risk of harm exceeds probable benefit or, more broadly, the added costs of the intervention do not provide proportional added benefits” [16]

**Table 2 healthcare-09-01693-t002:** Types of waste in imaging and potential consequences and general unintended consequences due to uncritical use of radiology.

Type	Explanation	Possible Consequences
Retake, reject	Image retaken, rejected, deleted or not used for diagnostic purposes, most often for quality reasons.	Unnecessary radiation exposure Increased examination time Discomfort related to the examination
Duplicate ordering	Duplicate imaging without changes in the patient’s state of health. No additional clinical utility.	
Repeated examinations at too short time interval	Repeat imaging without changes in the patient’s state of health or adequate observations. No additional clinical utility.	
Examinations ordered before patient examination and/or Imaging without sufficient clinical information	Makes it difficult to decide the appropriateness of imaging, to choose the right modality, as well as to interpret the image (reduced pre-test probability and positive predictive value).	Suboptimal imaging and (mis)interpretation May lead to retakes Delay in diagnostic and treatment courses
Screening examinations not supported by high-quality evidence	An examination routinely offered to a defined population for a certain problem or outside a screening program.	Generates overdiagnosis and potential overtreatment with related side effects
**General unintended consequences due to uncritical use of radiology**		
Overdiagnosis	Detection of a condition (from a true positive test result) that would not develop into symptoms or manifest disease during the person’s lifetime.	
Underdiagnosis	Suboptimal ordering and use of radiology may cause underdiagnosis, as the condition may not be visible due to inadequate imaging technique or misinterpretation, as the radiologist lacks necessary information.	
Incidental findings of no clinical relevance	Finding of a condition that (in some cases) can be clinically relevant when examining for something else (or when performing a “health check”).	

## Data Availability

Not applicable.

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
