# Peer review of "Visualizing the Invisible: Invisible Waste in Diagnostic Imaging"

_healthcare, 2021, doi:10.3390/healthcare9121693_

Round 1
Reviewer 1 Report
Summary
This review manuscript addresses the issue regarding invisible waste in diagnostic imaging in this era of filmless digital imaging. The authors discussed the invisible waste that are categorized by them, and concluded that visualizing the waste in diagnostic imaging and its ingrained drivers is important to improve the quality of the health care services.
Evaluation
The manuscript has been well written and reviewed. But, please correct mistype of does (p3, the 64 line) into dose.
I agree with the authors’ conclusion that visualizing the waste in diagnostic imaging and its ingrained drivers is important to improve the quality of the health care services. It is very important to inform to the readers that there has been many invisible waste in diagnostic imaging.

Author Response
Reviewer's comments are in black. Our responses are in red.
Reviewer 1's evaluation: I agree with the authors’ conclusion that visualizing the waste in diagnostic imaging is important to improve the quality of the health care services. It is very important to focus on the unnecessary imaging diagnosis such as CT or MRI and inform to the readers that there has been many invisible waste in diagnostic imaging.
OUR RESPONSE: We are most thankful for the thorough and constructive review, which has guided our revision and improved the manuscript.
My suggestions are as follows
1.Introduction
(a)Figure 1. showing waste films in bin can be deleted because the authors described and discussed in this digital era.
RESPONSE: Figure 1 has been removed.
(b) To describe concisely, last phrase (lines 47-51) can be deleted.
RESPONSE: The sentence “They are difficult to detect and measure, i.e., they are invisible” has been deleted. Please let us know if you wanted more to be deleted. This section now reads as follows: “The article starts by briefly presenting a short overview of different conceptions and types of waste in diagnostic imaging (in radiology). Then it investigates why these types of waste are so difficult to address. Thirdly, we identify and discuss external and internal drivers for wasteful imaging practice. Lastly, we use these findings to present specific measures to address wasteful imaging.”
- Waste in Imaging
- a) Please correct mistype of does (p3, the 64 line) into dose.
RESPONSE: “does” has been corrected to “dose.”
- Internal Drivers of Wasteful Imaging:Please give explanation of Type 1 error and Type 2 error (line 156-157)
RESPONSE: The sentence now reads: “In general, we tend to be more afraid of rejecting a true hypothesis (Type I error) than accepting a false hypothesis (Type II error), i.e., we are more afraid of ignoring than of over-doing.”
6.Discusssion
- a) Please delete the 1st phrase, or describe in Introduction.
RESPONSE: We have rewritten the first section of the discussion but are not sure if we address what the reviewer had in mind.
- b) Please describe to focus on what benefits may occur on the health care services after decreasing invisible wasting imaging
RESPONSE: This is a very good and important point. We now have added the following: “By visualizing invisible waste in imaging, and knowing its drivers, we are much better equipped at addressing this problem. In particular, we must avoid duplicates, reduce retakes, extend time interval for repeated examinations (where appropriate), halt examinations without sufficient information, stop low value imaging including screening with poor evidence to avoid overdiagnosis and overtreatment, as well as being cautious with respect to incidental findings without or uncertain significance. By reducing waste, we can improve the quality of care by increasing benefit and reducing harm. Moreover, we can promote efficiency and justice as resources from low value care are freed to provide high value care. Many measures are available (21), but it is crucial that they are adapted to the specific context.”
ï¼—.Conclusion
- a) Please delete the 1st sentence.
RESPONSE: We are not quite sure if understand this suggestion, as deleting the first sentence would make the first section incomplete. We have rewritten this section, but please give us further advice, it this is not what you had in mind.
References: Reference 15: please fill out “year”,”volume” and “page”
RESPONSE: This is a submitted manuscript, so we cannot fill in the details yet.
Reviewer 2 Report
The topic covered by the authors in the paper is interesting, well presnted in form and methodology and has some proejections for future reserch which are included explicitly in the conclusions section. Moreover, the paper is easy to read and is well documented. Therefore, my conclusion is the following: the actual version of the paper can be accepted and published by the Healthcare journal.

Author Response
Reviewer's comments are in black. Our responses are in red.
We are most thankful for the thorough and constructive review, which has guided our revision and improved the manuscript.
In a broad sense, the paper is well written and has a significant contribution to the topic. However, the current version of the paper can be improved in the following two weak issues:
(a) The abstract must be rewritten. The abstract is strange. Probably is written in a format of other journal. My suggestion is to consider a new abstract following the current form of the articles published by Healthcare and following the suggestions of some methodologies to write abstracts, for instance: - Andrade C. How to write a good abstract for a scientific paper or conference presentation. Indian J Psychiatry. 2011; 53(2):172-175. doi:10.4103/0019-5545.82558.
RESPONSE: We have now rewritten the abstract as follows: "Abstract: There is extensive waste in diagnostic imaging, at the same time as there are long waiting lists. While the problem of waste in diagnostics has been known for a long time, the problem persists. Accordingly, the objective of this study is to investigate various types of waste in imaging and why they are so pervasive and persistent in today’s health services. After a short overview of different conceptions and types of waste in diagnostic imaging (in radiology) we identify two reasons why these types of waste are so difficult to address: 1) they are invisible in the health care system, and 2) wasteful imaging is driven by strong external forces and internal drivers. Lastly, we present specific measures to address wasteful imaging. Visualizing and identifying the waste in diagnostic imaging and its ingrained drivers is one important way to improve the quality and efficiency of the health care services.”
However, we were somewhat bewildered as we tried to follow the instructions for authors that say that: “The abstract should be a total of about 200 words maximum. The abstract should be a single paragraph and should follow the style of structured abstracts, but without headings: 1) Background: Place the question addressed in a broad context and highlight the purpose of the study; 2) Methods: Describe briefly the main methods or treatments applied. Include any relevant preregistration numbers, and species and strains of any animals used. 3) Results: Summarize the article's main findings; and 4) Conclusion: Indicate the main conclusions or interpretations. The abstract should be an objective representation of the article: it must not contain results which are not presented and substantiated in the main text and should not exaggerate the main conclusions.” We also found examples where this format was use, but please guide us to improve the abstract.
(b) Section 2: In section the authors present a list of references with the concept and types of “waste in imaging”. However they does not present the methodology which was applied to select the references. Then, my suggestion is to include a paragraph with the methodology which was applied too select the references.
RESPONSE: To address this issue, we have added this to the discussion: “This article is not a systematic review, and it is neither exhaustive nor exclusive concerning available literature. However, many references stem from one scoping review (15) and a systematic review (21) which is supplemented wither references found by snowballing. Nonetheless, this study can be of great value for further research including systematic reviews of waste in imaging.”
Conclusion: The results obtained in the paper are new and interesting, However, some minor inaccuracies are necessary to be fixed. After the revision I will have no objection to recommend it for publication in Healthcare.
RESPONSE: We are most thankful for the encouraging review and hope that we have been able to address the issues in a satisfactory manner.
Reviewer 3 Report
The issue raised in the article is significant when it comes to health care and its management. Cost optimization, reducing the cost of research per patient, has been a problem of medical systems in many countries for years. The authors point out that waste after the form of waste from radiological imaging is one of the elements that can affect this.
The article does a good job of showing what types of radiological waste there are, the reasons they are created, and the implications of their creation for healthcare and the patient. However, I miss both the literature review and the authors' own conclusions on how to address this problem. I would ask you to address this issue in your discussion and conclusion.
Author Response
We are most thankful for the thorough and constructive review, which has guided our revision and improved the manuscript.
The reviewer's comments are in black, while our responses are in red.
The issue raised in the article is significant when it comes to health care and its management. Cost optimization, reducing the cost of research per patient, has been a problem of medical systems in many countries for years. The authors point out that waste after the form of waste from radiological imaging is one of the elements that can affect this.
The article does a good job of showing what types of radiological waste there are, the reasons they are created, and the implications of their creation for healthcare and the patient. However, I miss both the literature review and the authors' own conclusions on how to address this problem. I would ask you to address this issue in your discussion and conclusion.
RESPONSE: This is a very good and important point. We now have added the following to the discussion: “By visualizing invisible waste in imaging, and knowing its drivers, we are much better equipped at addressing this problem. In particular, we must avoid duplicates, reduce retakes, extend time interval for repeated examinations (where appropriate), halt examinations without sufficient information, stop low value imaging including screening with poor evidence to avoid overdiagnosis and overtreatment, as well as being cautious with respect to incidental findings without or uncertain significance. By reducing waste, we can improve the quality of care by increasing benefit and reducing harm. Moreover, we can promote efficiency and justice as resources from low value care are freed to provide high value care. Many measures are available (21), but it is crucial that they are adapted to the specific context.” Moreover, the following is added to the conclusion: “To reduce waste in imaging, we must avoid duplicates, reduce retakes and repeated examinations, halt examinations without sufficient information, stop low value examinations and screening with poor evidence, as well as looking for incidental findings of uncertain significance.”